# PRIORITIZED OFFLINE GOAL-SWAPPING EXPERIENCE REPLAY

**Wenyan Yang**[1]**, Joni Pajarinen**[2]**, Dingding Cai**[1]**, and Joni-Kristian Kämäräinen**[1]
[1]Computing Sciences, Tampere University, Finland,
[2]Department of Electrical Engineering and Automation, Aalto University, Finland
`first.surname@{tuni.fi,aalto.fi}`

## ABSTRACT

In goal-conditioned offline reinforcement learning, an agent learns from previously collected data to go to an arbitrary goal. Since the offline data only contains a finite number of trajectories a main challenge is how to generate more data. Goal-swapping generates additional data by switching trajectory goals but while doing so produces a large number of invalid trajectories. To address this issue we propose prioritized goal-swapping experience replay (PGSER). PGSER uses a pre-trained Q function to assign higher priority weights to goal-swapped transitions that allow reaching the goal. In experiments, PGSER significantly improves over baselines in a wide range of benchmark tasks.

## 1 INTRODUCTION

Reinforcement learning (RL) has been used to great success in a variety of tasks, from gaming Mnih et al. (2013); Frazier & Riedl (2019); Mao et al. (2022) to robotics Brunke et al. (2022); Nguyen & La (2019). Typically RL is used in a setting where the agent learns to optimize behavior for one specific task. To allow for a more general RL solution, goal-conditioned RL (GCRL) Liu et al. (2022b); Chane-Sane et al. (2021); Andrychowicz et al. (2017) learns a policy that can reach arbitrary goals without needing to be retrained. Training GCRL agents can be difficult due to the sparsity of rewards in GCRL tasks, forcing the agent to explore the environment, which can be unfeasible or even dangerous in some real-world tasks. To utilize RL without environment interactions offline RL allows learning a policy from a dataset without putting real environments at risk Levine et al. (2020); Prudencio et al. (2022). Offline goal-conditioned RL (offline GCRL) combines the generalizability of GCRL and the data-efficiency of offline RL, making it a promising approach for real-world applications Ma et al. (2022).

Although offline goal-conditioned reinforcement learning (GCRL) is an appealing concept, it faces some challenges. The first is that the offline dataset only covers a limited state-goal-action space, which can cause incorrect value function estimations for out-of-distribution observations. This can lead to compounding errors in policy deviation Levine et al. (2020); Prudencio et al. (2022). Additionally, each state can have multiple goals, making it hard to learn from a limited state-goal observation space Chebotar et al. (2021). To deal with this issue, prior works have applied hindsight labeling to generate goal-conditioned observations in sub-sequences Ghosh et al. (2019); Yang et al. (2022); Ma et al. (2022); Andrychowicz et al. (2017). However, this often leads to overfitted goal-conditioned policies Chebotar et al. (2021). Furthermore, the goal-chaining technique proposed by Chebotar et al. (2021) is not able to handle noisy data properly, while inefficiently swapping goals Ma et al. (2022). In order to achieve successful skill learning across multiple trajectories, a solution is needed that can make agents effectively learn from the limited data.

This paper presents Prioritized Goal-Swapping Experience Replay (PGSER), an approach for offline Goal-Conditioned Reinforcement Learning (GCRL) tasks. PGSER provides two main benefits: (1) allowing the agent to learn goal-conditioned skills across different trajectories, and (2) maximizing offline data utilization. The process of PGSER is illustrated in Figure 1. During the offline training stage, random goal-swapping augmentation is used to generate new goal-conditioned transitions $\zeta_{aug}$; a pre-trained Q function is then used to estimate the priority of each $\zeta_{aug}$, and these transitions are stored in an additional prioritized experience replay buffer $\beta_{aug}$. During training, data is sampled

from both the original dataset buffer $\beta$ and the added buffer $\beta_{aug}$, which helps to improve the accuracy and effectiveness of the training process and enables the agent to learn goal-conditioned skills across different trajectories. We evaluated PGSER on a wide set of offline GCRL benchmark tasks. The experimental results show that PGSER outperforms baselines.

Figure 1: An illustration of prioritized goal-swapping experience replay. During the offline training stage, we conduct random goal-swapping augmentation to create new goal-conditioned transitions $\zeta_{aug}$. A pre-trained Q function estimates the priority $w$ of corresponding $\zeta_{aug}$ in order to increase the priority of augmented transitions which are more likely to reach the goal. We store both $\zeta_{aug}$ and $w$ into an additional prioritized experience replay buffer $\beta_{aug}$. During training the final policy data will be sampled from both the original dataset buffer $\beta$ and the prioritized goal-swapping buffer $\beta_{aug}$ for goal-conditioned skill learning.

## 2 PRELIMINARIES

**Goal-conditioned Markov decision process.** The classical Markov decision process (MDP) $\mathcal{M}$ is defined as a tuple $< \mathcal{S}, \mathcal{A}, \mathcal{T}, r, \gamma, \rho_0 >$, where $\mathcal{S}$ and $\mathcal{A}$ denote the state space and the action space, $\rho_0$ represents the initial states' distribution, $r$ is the reward, $\gamma$ is the discount factor, and $\mathcal{T}$ denotes the state transition function Sutton & Barto (2018). For goal-conditioned tasks, an additional vector $g$ specifying the desired goal is included. This augmentation of MDP is referred as the goal-conditioned MDP (GC-MDP), $< \mathcal{S}, \mathcal{G}, \mathcal{A}, \mathcal{T}, r, \gamma, \rho_0, p_g >$ Liu et al. (2022b). This GC-MDP includes a goal space $\mathcal{G}$ and a goal distribution $p_g$, as well as a tractable mapping function $\phi : \mathcal{S} \to \mathcal{G}$ to map the state to the corresponding goal. The state-goal pair $(s, g)$ forms a new observation, which is used as the input for the agent $\pi(a|s, g)$. The objective of GC-MDP can be formulated as:

$$\mathcal{J}(\pi) = \mathop{\mathbb{E}}_{a_t \sim \pi(\cdot|s_t,g), g \sim p_g, s_{t+1} \sim \mathcal{T}(\cdot|s_t,a_t)} \left[ \sum_{t=0}^{\infty} \gamma^t r_t \right]. \tag{1}$$

Two value functions are defined to represent the expected cumulative return in a goal-conditioned Markov Decision Process (GC-MDP): a state-action value $Q$ and a state value $V$. The $V^\pi(s, g)$ function is the goal-conditioned expected total discounted return from the observation pair $(s, g)$ using policy $\pi$, while the $Q^\pi(s, a, g)$ function estimates the expected return of an observation $(s, g)$ for the action $at$ for the policy $\pi$. Additionally, the advantage function $A^\pi(s, g, a)$ is another version of the Q-value with lower variance, defined as $A^\pi(s, g, a) = Q^\pi(s, g, a) - V^\pi(s, g)$. When the optimal policy $\pi^*$ is obtained, the two value functions converge to the same point, i.e. $Q^*(s, g, a) = V^*(s, g)$ Sutton & Barto (2018). In this work, we define a sparse reward

$$r(s, g) = \begin{cases} 0, & \text{if } ||\phi(s), g|| < \epsilon \\ -1, & \text{otherwise} \end{cases}, \tag{2}$$

where $||\phi(s), g||$ is a distance metric measurement, and $\epsilon$ is a distance threshold. We set the discount factor $\gamma = 1$. In such case, $V(s, g)$ represents the expected horizon from state $s$ to goal $g$, and

$Q(s, g, a)$ represents the expected horizon from state $s$ to the goal $g$ if an action $a$ is taken. This setting yields an intuitive objective: finding the policy that takes the minimum number of steps to achieve the task's goal. This setting produces an intuitive objective: *Find the policy that takes the minimum number of steps to achieve the task's goal.*

**Offline goal-conditioned reinforcement learning.** In offline reinforcement learning (offline RL), an agent must work with a set of static, pre-existing data rather than interacting with an environment to collect data. This data is often collected by unknown policies Levine et al. (2020); Prudencio et al. (2022). In the offline goal-conditioned setting, the objective is the same as in online goal-conditioned RL, as defined by Equation 1. The offline data consists of goal-conditioned trajectories $\zeta$ stored in a dataset $\mathcal{D} := \zeta_{i,i=1}^{N}$, where $N$ is the number of stored trajectories and

$$\zeta_i = \{< s_0^i, \eta_0^i, a_0^i, r_0^i >, < s_1^i, \eta_1^i, a_1^i, r_1^i >, ..., < s_T^i, \eta_T^i, a_T^i, r_T^i >, g^i\} .$$

$\eta$ is the state's corresponding goal representation calculated using $\eta_t = \phi(s_t)$. The task goal $g^i$ is randomly sampled from $p_g$ and the initial state $s_0 \sim \rho_0$. Note that some trajectories can be unsuccessful ($\eta_T^i \neq g^i$).

**Prioritized experience replay.** Prioritized experience replay (PER) Schaul et al. (2015) allows reinforcement learning from a diverse set of experiences. PER uses the temporal difference (TD)-error of each experience to determine the priority of that experience in the replay buffer. By prioritizing experiences with higher TD-errors, the agent can focus on those experiences that are most likely to reinforce its learning. This technique can be used to improve the convergence of reinforcement learning algorithms, allowing them to learn more effectively.

## 3 GENERALIZATION PROBLEM OF GOAL-CONDITIONED RL

Goal-Conditioned Reinforcement Learning (GCRL) aims to learn a general policy that can reach arbitrarily goals Liu et al. (2022b). However, the offline dataset state-goal pairs only cover a limited space of the goal-conditioned MDP. In other words, if we train a policy with the offline dataset, the policy learns to reach goals within a single trajectory in the dataset. This solution is not general but undesirably specific. Fig. 2 shows a simple visual example. In Fig. 2, each color represents a goal-conditioned trajectory. If we use the dataset in Fig. 2 for offline RL, we eventually get an overfitted policy that can only achieve a single goal starting from a given state. Ideally, we want an agent that can achieve as many goals as possible (the blue trajectory).

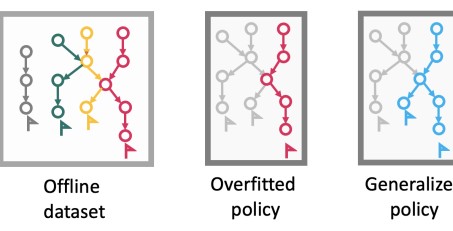

Offline      Overfitted      Generalized
dataset       policy        policy

Figure 2: The generalizability problem of offline GCRL: the agent needs to learn how to reach a goal from a state without a trajectory for this state-goal pair. Each color represents an individual goal-conditioned trajectory.

Several techniques have been proposed to learn a generalized and effective goal-conditioned policy. Actionable Models (AM) Chebotar et al. (2021) use a goal-chaining technique that augments the dataset by assigning conservative values to the augmented out-of-distribution data for Q-learning. However, the performance of AM is limited when the dataset contains noisy data labels Ma et al. (2022). Hindsight Experience Replay (HER) Andrychowicz et al. (2017); Yang et al. (2022) relabels trajectory goals to states that were actually achieved instead of the task-commanded goals, efficiently utilizing the data within a single trajectory. However, HER is not able to connect different trajectories as goal-chaining does. Contrastive Reinforcement Learning (CRL) Eysenbach et al. (2022) conducts contrastive learning on the goal-conditioned task by swapping the future observations between different goal-conditioned trajectories to generalize the skills. However, CRL focuses more on representation learning rather than addressing the generalizability of offline goal-conditioned tasks.

## 4 METHOD

To solve the challenges discussed in the previous section, we propose reusing the pre-trained Q function to generate better goal-swapping augmentation samples. A schematic illustration of the

method inside the RL framework is in Fig. 1. We first conduct goal-swapping data augmentation to generate new state-goal pairs for Q value estimation. In the second stage, we use the pre-trained Q function to filter out the low-quality augmented samples and store the reachable samples for agent learning. In this work, we assume that in the same environment, all goals are reachable from all states. Our approach can be used with off-policy offline RL methods (e.g., TD3BC Fujimoto & Gu (2021), etc.), which aim to approximate the optimal Q value from the offline dataset.

## 4.1 GOAL-SWAPPING DATA AUGMENTATION

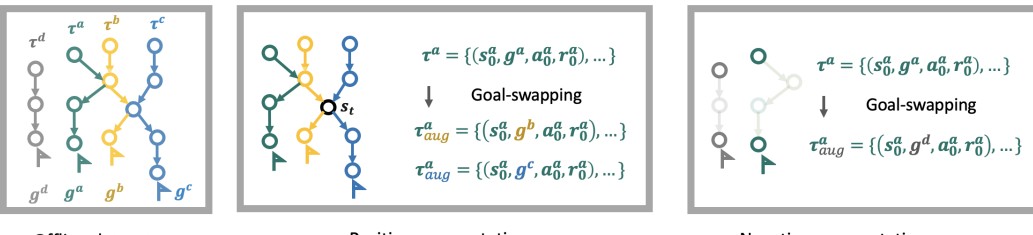

Figure 3: An example of goal-swapping data augmentation. (a) Original offline trajectories (denoted by four different colors). (b) Positive augmentation example where the goals are reachable in each generated trajectory. (c) shows the negative augmentation, where the goals are no longer reachable.

As discussed in Section 3, the offline goal-conditioned dataset contains only a limited number of state-goal observations. Therefore, training with offline data often results in overfitted policies. An illustrative example is in Figure 3(a). In this example environment, the goals $g^a$ and $g^b$ are (forward) reachable from three states $s_0^a$, $s_0^b$, and $s_0^c$. However, the original conditioned trajectories $\tau^a$, $\tau^b$, and $\tau^c$ limit the agent's exploration to the known state-goal pairs $s_0^a \to g^a$, $s_0^b \to g^b$, and $s_0^c \to g^c$.

To maximize the efficiency of our agent, we propose a goal-swapping augmented experience replay technique (detailed in Algorithm 1). This technique randomly samples two trajectories and swaps their goals, creating a virtually infinite amount of new goal-conditioned trajectories. As reinforcement learning is a dynamic programming method, this allows us to connect the state-goal pairs across different trajectories. By doing so, we are able to expand the range of achievable goals significantly while also increasing the speed and accuracy of our agent's learning process.

Let's continue the example in Figure 3(a), where the goals $g \in [g^a, g^b, g^c]$ are reachable from the states $s \in [s_0^a, s_0^b, s_0^c]$ although they are not explicitly present in the offline dataset. However, if the goals are swapped between the original three trajectories $[\tau^a, \tau^b, \tau^c]$, the augmented goal-conditioned tuples shown in Figure 3(b) become available. The Q-learning-based approaches can leverage dynamic programming to chain the new trajectories and backpropagate their state-goal (state-goal-action) values to the previous pairs. An illustration of this can be seen in Fig.3(b), where the state $s_t$ is the "hub state" shared by the three original trajectories and from which all goals $g^i \sim [g^a, g^b, g^c]$ are reachable. The values of these goal-conditioned states, $Q(s_t, a_t^i, g^i)$, can be estimated and recursively backpropagated to $Q(s_0, a_0^i, g^i), i \in [a, b, c]$.

The purpose of goal-swapping augmentation is to create as many state-goal pairs as possible so that the Q-learning (temporal difference learning style) can backpropagate values over all trajectory combinations. Alg. 1 provides an illustration of how this is done. By swapping goals between trajectories, the offline RL methods can extend the dataset and make use of diverse information. This ensures that the agent can explore and discover more accurate and robust policies.

## 4.2 REACHABLE TRANSITIONS IDENTIFICATION

Although temporal-difference learning can connect goals across trajectories, applying the goal-swapping augmentation technique can be tricky. The reason is that the goal-swapping process is random, creating many non-optimal state-action pairs. Those augmented state-action pairs may not even have a solution (the augmented goals are not reachable) within the offline dataset. This is demonstrated in Fig. 3(c). To alleviate the inefficiency problem of random data augmentation, we

---

**Algorithm 1** Random goal-swapping experience replay

---

**Require:** Denote the dataset as $D$, goal-conditioned tuples as $\zeta$, state-goal mapping function $g_i = \phi(s_i)$, and reward function $r = R(\phi(s), g)$.
1: Sample goal-conditioned transitions $\zeta$ from $D$:
$\quad\quad \zeta_i = \{g, s, a, r, s'\} \sim D$.
2: Sample random goals: $g_{rand} \sim D$
3: Generate $\tau_{rand}$ by replacing $g$ with $g_{rand}$ in $\zeta$:
$\quad\quad \zeta_{aug} = \{g_{rand}, s, a, r_{aug}, s'\}$
$\quad\quad$ where $r_{aug} = R(\phi(s'), g_{rand})$
4: Return $\zeta$ and $\zeta_{aug}$

---

aim to design a mechanism that makes the agent remember the reachable goal-conditioned transitions (positive samples) and ignore the unreachable ones (negative samples). *In this work, we use a pre-trained Q-function as a "past-life" agent to improve the efficiency of random goal-swapping.*

Based on Eq.1, the Q value represents the expected horizon to reach the goal when $\gamma = 1$. Denote the maximum horizon of the task as $H$, the reachable goal as $g$, and the unreachable goal as $g'$. The Q value estimation should result in $Q(s_0, g, a) \geq -H$, indicating that $g$ is reachable, and $Q(s_0, g', a) < -H$ for an unreachable goal. As is shown in Fig. 4, given a set of observed deterministic MDP transitions, the optimal state-goal-action value $Q^*(s, g, a) = -2$ and the unreachable state-goal-action value $Q(s, g', a) = -4$. Such a simple example implies that the trained Q function can be used as an identifier to tell if the goal is reachable for a given state-goal-action pair.

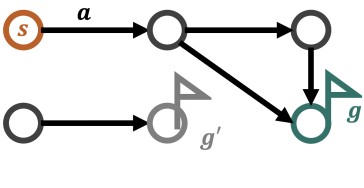

$Q^*(s, g, a) = -2 > -H_{max}$
$Q(s, g', a) = -4 = -H_{max}$

Figure 4: Consider a deterministic MDP, the reward and discount factor is defined in Sec 2, and the maximum horizon $H_{max} = 4$. The agent is trained with Q-learning with above transitions.

Now, we consider the random goal-swapping data augmentation process, denoting transitions in successful trajectories (goals that are reached) as positive transitions $\zeta_P = \{s_P, g_P, a_P, r_P, s'_P\}$, and random goal-swapping augmentation as negative training samples $\zeta_N = \{s_N, g_N, a_N, r_N, s'_N\}$ (generated using Alg. 1). We also have a pretrained value function $\mathcal{Q}$. For this data, $\mathcal{Q}$ will assign higher values to $\zeta_P$ and, likely, lower values to $\zeta_N$ when goals are not reachable. Consequently, this pre-trained $\mathcal{Q}$ can be used to identify how likely a goal-conditioned tuple $\{s, g, a, r, s'\}$ belongs to a reachable trajectory. In other words, the smaller the $\{s, g, a, r, s'\}$ q-value is, the less likely its goal is reachable. We provide estimated Q-value distributions in Sec. 5.2 to further illustrate this concept.

### 4.3 PRIORITIZED GOAL-SWAPPING AUGMENTATION

The random goal-swapping augmentation creates goal-swapped transitions at the same frequency, regardless of if the augmentation is positive or negative. To improve the efficiency of the augmentation, we introduce an additional experience replay buffer. This buffer is used to remember positive augmentations during offline training. As is discussed above, the Q function can be naturally used to identify if a state-goal-action is a goal-reachable observation. To assign priority values to the augmented transitions, we pre-train a value function, $\mathcal{Q}$ with Q-learning-based offline RL methods. After the pretraining, this function $\mathcal{Q}$ can estimate how likely the augmentations are positive. The higher the estimated Q value, the more likely the augmented transition is a positive augmentation, and it shall be sampled more frequently. We illustrate the process of creating an experience replay buffer in Fig. 1.

The additional experience replay buffer $\beta_{aug}$ is implemented in a PER style. $\beta_{aug}$ stores the goal-swapping augmented transitions $\zeta_{aug} = \{s, g_{rand}, a, r, s'\}$ and its priority value $\mathcal{Q}(s, g_{rand}, a)$. During the training, the agent samples the goal-swapped augmented transitions according to the their priority values. We illustrate our framework in Fig. 1 and provide a pseudo-code in Alg. 2. In this way, the positive goal-swapped augmentations will be sampled more frequently than the negative

ones. In this way, we have a mechanism that can guide the agent efficiently use the goal-swapping augmentation to learn general goal-conditioned skills from the offline dataset.

---

**Algorithm 2** Prioritized goal-swapping experience replay

---

**Require:** The original offline buffer as $\beta$, the augmented buffer as $\beta_{aug}$, goal-conditioned tuples as $\zeta$, state-goal mapping function $g_i = \phi(s_i)$, reward function $r = R(\phi(s), g)$, a pre-trained Q-function $\mathcal{Q}$
 1: Pre-train a off-policy value function $\mathcal{Q}(s, g, a)$ using random augmentation (Alg. 1).
 2: Fill-in $\beta_{aug}$ using $\mathcal{Q}$:
     a) Generate $\zeta_{aug} = \{g_{rand}, s, a, r_{aug}, s'\}$ using Alg. 1.
     b) Estimate priority weight $w = \mathcal{Q}(s, g_{aug}, a)$.
     c) Store $\zeta_{aug}$ into $\beta_{aug}$ and set its sampling priority using $w$.
 3: Re-training the RL agent:
     a) Sample $\zeta \sim \beta$, Sample $\zeta_{aug} \sim \beta_{aug}$.
     b) Combine $\zeta$ and $\zeta_{aug}$ for offline RL training.

---

## 5 EXPERIMENTS

The experiments in this section were designed to verify our main claims: 1) The pre-trained value function is able to identify if the goal is reachable; 2) The filtered goal-swapping experience replay can improve offline GCRL performances.

**Tasks.** Six goal-conditioned tasks from Plappert et al. (2018) were selected for the experiments. The tasks include four fetching manipulation tasks (FetchReach, FetchPickAndPlace, FetchPush, FetchSlide), and two dexterous in-hand manipulation tasks (HandBlock-Z, HandEgg). In the fetching tasks, the virtual robot should move an object to a specific position in the virtual space. In the dexterous in-hand manipulation tasks, the agent is asked to rotate the object to a specific pose. The offline dataset for each task is a mixture of 500 expert trajectories and 2,000 random trajectories. The fetching tasks and datasets are taken from Yang et al. (2022). The expert trajectories for the in-hand manipulation tasks are generated similarly to Liu et al. (2022a).

**Baselines.** For the experiments, we use TD3 Fujimoto et al. (2018) and TD3BC Fujimoto & Gu (2021) as they are built on the classical Q-learning approaches and solid baselines for many offline RL tasks Levine et al. (2020). The HER ratio of 0.5 was used with all methods in all experiments. We trained each method for 10 seeds, and each training run uses 500k updates. The mini-batch size is 512. We used the cumulative test rewards from the environments as the performance metric in all experiments.

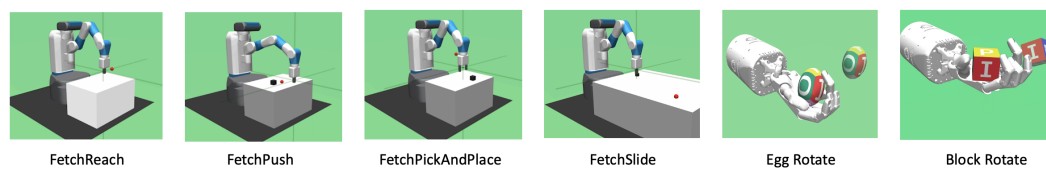

FetchReach      FetchPush      FetchPickAndPlace      FetchSlide      Egg Rotate      Block Rotate

Figure 5: Six goal-conditioned tasks.

### 5.1 PRE-TRAINING VALUE FUNCTION

To construct the prioritized goal-swapping experience replay, we first need to train a Q-function that can assign reasonable values to state-goal-action pairs. In this stage, several algorithms can be chosen to pre-train the value function, such as conservative Q-learning Kumar et al. (2020), actionable models Chebotar et al. (2021), TD3BC Fujimoto & Gu (2021), etc. In this work, we choose to use TD3BC Fujimoto & Gu (2021) as it has a minimal modification on the original Q-learning (its value function is not heavily modified as other offline methods). To ensure the Q function covers as much state-goal-action space as possible, we combined the random goal-swapping augmentation

(Alg. 1) with TD3BC during the pre-training. We trained each value function for each task with 1000k updates, and each update's batch size is 512.

## 5.2 PRE-TRAINED Q VALUE DISTRIBUTION

We aim to validate our first research proposal: whether a pre-trained Q function can identify if transitions are goal-reachable. To do this, we created two classes of data: $\zeta_N$ and $\zeta_P$. $\zeta_N$ contains randomly augmented transitions with goal-swapping, while $\zeta_P$ contains expert transitions. We used a pre-trained value function $\mathcal{Q}$ to evaluate all transitions in these two sets, and the results are displayed in Fig. 6. We believe that the differences in the Q-value distributions provide evidence to support our hypothesis that a pre-trained Q function can identify if transitions are goal-reachable.

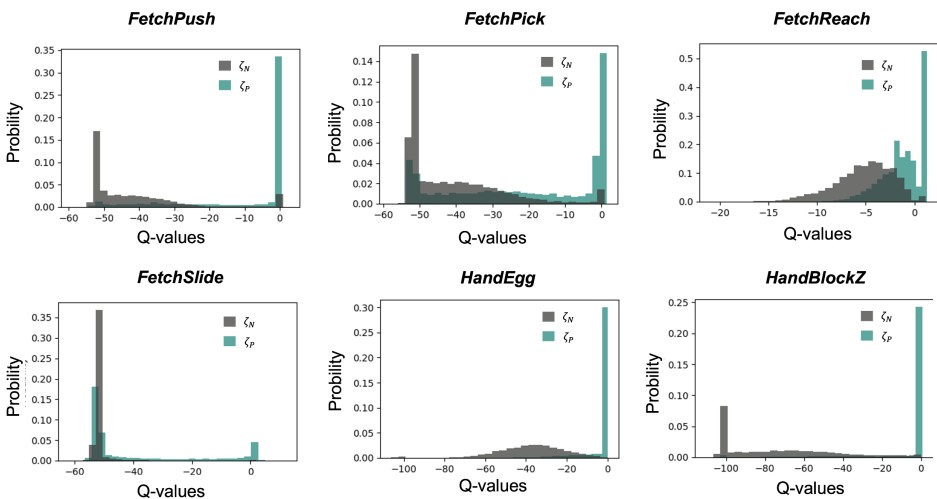

Figure 6: The histogram of Q-value distributions. In this figure we compare the Q-value distributions of goal-swapping augmentations (dark gray) and the positive goal-conditioned samples (dark green).

Overall, the Q-value distribution of $\zeta_P$ notably differs from that of $\zeta_N$. Generally, $\zeta_P$ has higher Q-values than $\zeta_N$, which validates our proposal in Section 4.2, that goal-reachable transitions will possess higher Q-values. This statement holds true for the tasks FetchPush, FetchPick, HandEgg and HandBlockZ. It is worth noting that for the simple task FetchReach, the Q-value of $\zeta_N$ has a similar distribution as $\zeta_P$, suggesting that the random goal-swapping augmentation likely yields positive augmentations. Moreover, in task FetchSlide, the expert also provides unsuccessful trajectories, resulting in $\zeta_P$ exhibiting two peaks in its Q-value (one on the minimal Q-value side and one on the maximum Q-value side). This is reflected in the Q-value distribution of $\zeta_N$, which mirrors the distribution of $\zeta_P$.

We also implement a set of simple linear classifiers that utilize the estimated Q-value to classify whether a given state-goal-action pair $\zeta = \{s, g, a, r, s'\}$ is a reachable transition. The classification results are presented in Table 1. As shown in the table, the pre-trained value function $\mathcal{Q}$ is able to accurately estimate the goal-reachable transitions in the FetchPush, FetchPick, HandEgg, and HandBlockZ tasks. The results demonstrate that the pre-trained value function $\mathcal{Q}$ is able to identify the goal-reachable transitions.

## 5.3 PERFORMANCE

In this experiment, we aim to study the impact of prioritized goal-swapping experience replay on offline goal-conditioned reinforcement learning (RL). We used two deep Q-learning approaches, TD3 and TD3BC, to compare the performance of agents trained with prioritized goal-swapping experience replay (Alg. 2) versus the original algorithm and agents trained with random goal-swapping experience replay (Alg. 1). The comparison results are presented in Tab. 2.

Overall, the prioritized goal-swapping experience replay provided more robust performance improvements than the random goal-swapping augmentation for offline GCRL tasks. In particular, the

Table 1: The goal-reachable identification accuracy. We use the estimated Q-value as the classification feature to identify if the transition belongs to a successful trajectory (goals are reachable).

|  | FetchPush | FetchSlide | FetchPick | FetchReach | HandBlockZ | HandEggRotate |
|---|---|---|---|---|---|---|
| **Logistic regression** | 0.915 | 0.970 | 0.925 | 0.770 | 0.880 | 0.905 |
| **KNN** | 0.920 | 0.965 | 0.910 | 0.785 | 0.885 | 0.905 |
| **Naive Bayesian** | 0.920 | 0.965 | 0.905 | 0.770 | 0.890 | 0.915 |
| **SVM** | 0.920 | 0.965 | 0.905 | 0.775 | 0.885 | 0.905 |

Table 2: Evaluation of the goal-swapping augmentation in offline RL. The tested methods were trained without (*method*) and with (*method*-aug) the *random* goal-swapping augmentation (Alg. 1). The methods (*method*-mem) were trained with *prioritized* goal-swapping augmentation. The performance metric is the average cumulative reward over 50 random episodes. ▨ indicates that the augmented variant outperforms the original baseline according to the t-test with p-value $< 0.05$. **text** indicates that the (*method*-mem) variant outperforms the (*method*-aug) variant according to the t-test with p-value $< 0.05$.

| Algorithms | FetchPush | FetchSlide | FetchPick | FetchReach | HandBlockZ | HandEggRotate |
|---|---|---|---|---|---|---|
| TD3BC-mem | **$29.85 \pm 11.57$** | $1.23 \pm 2.85$ | $28.74 \pm 15.55$ | $48.12 \pm 1.37$ | $28.43 \pm 30.13$ | **$34.07 \pm 32.57$** |
| TD3BC-swap | $27.52 \pm 16.27$ | $0.58 \pm 1.48$ | $27.38 \pm 12.86$ | $47.35 \pm 1.94$ | $28.32 \pm 27.81$ | $32.12 \pm 37.03$ |
| TD3BC (baseline) | $26.94 \pm 14.42$ | $1.21 \pm 3.98$ | $26.36 \pm 15.75$ | $47.72 \pm 1.12$ | $16.35 \pm 29.78$ | $21.56 \pm 32.68$ |
| TD3-mem | **$32.25 \pm 13.89$** | $1.73 \pm 2.39$ | $23.66 \pm 13.78$ | $48.22 \pm 3.87$ | **$24.57 \pm 20.13$** | **$15.83 \pm 6.27$** |
| TD3-swap | $27.83 \pm 15.47$ | $0.99 \pm 1.83$ | $24.23 \pm 15.05$ | $47.09 \pm 1.285$ | $15.43 \pm 27.23$ | $3.28 \pm 2.46$ |
| TD3 (baseline) | $20.68 \pm 16.27$ | $1.01 \pm 2.96$ | $18.26 \pm 8.43$ | $46.52 \pm 1.47$ | $13.25 \pm 21.08$ | $7.32 \pm 5.27$ |

random goal-swapping augmentation even caused a decrease in performance for the TD3 agent on the HandEggRotate task. However, for the FetchSlide task, neither random nor prioritized goal-swapping augmentation resulted in performance improvements. This could be attributed to the limited number of successful trajectories in the offline dataset, which limits the agent's performance.

In addition, we observed that the prioritized goal-swapping experience replay even improved the performance of the non-offline RL agent TD3, making it comparable to TD3BC. This suggests that the prioritized goal-swapping augmentation can effectively cover as much state-goal-action space as possible, allowing the Q-function to be trained appropriately.

# 6 CONCLUSIONS

In this paper, we proposed a novel and innovative approach to offline goal-conditioned reinforcement learning using a prioritized goal-swapping technique. Our method employs a pre-trained agent to generate meaningful and effective data augmentations that can help the agent acquire general skills from the offline dataset. Compared to random goal-swapping data augmentation, our method has proven more robust and successful in improving performance.

One limitation of this work is its focus on goal-conditioned reinforcement learning (RL) problems. Although the idea of using pre-trained value functions to assign sampling weights could be applied to more traditional offline RL tasks, training an additional value function is still necessary, which can be time-consuming. To address this, future work should focus on developing more general and efficient techniques in terms of training.

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
