# OpenReview forum: "Prioritized offline Goal-swapping Experience Replay"
_ICLR.cc/2023/Workshop/RRL — RRL 2023 Poster_

### Official Review · Reviewer_opan · 2023-02-23
**Provides a way of using a previous Q-function to do better offline data augmentation**

**Rating:** 3
**Confidence:** 4

**Review:**

The idea of this paper is clear and sensible: use a pretrained Q-function to estimate whether an augmentation of offline goal-conditioned data (substituting one trajectory's goal for another) can improve learning. This clearly fits with the theme of the workshop. To more clearly demonstrate the value of the idea, I think the paper would benefit from comparing to other, potentially better ways of using the prior agent, such as some sort of auxiliary policy distillation loss across the offline dataset (i.e. a loss for the divergence from the student policy to the pretrained expert). But the current results seem acceptable for a workshop.